# Recognition of flight cadets brain functional magnetic resonance imaging data based on machine learning analysis

Lu Ye, Shuhao Weng, Dongfeng Yan *, Shan Ma, Xi Chen

Institute of Flight Technology, Civil Aviation Flight University of China, Guanghan, China

* yandongfeng@cafuc.edu.cn

## Abstract

The rapid advancement of the civil aviation industry has attracted significant attention to research on pilots. However, the brain changes experienced by flight cadets following their training remain, to some extent, an unexplored territory compared to those of the general population. The aim of this study was to examine the impact of flight training on brain function by employing machine learning(ML) techniques. We collected resting-state functional magnetic resonance imaging (resting-state fMRI) data from 79 flight cadets and ground program cadets, extracting blood oxygenation level-dependent (BOLD) signal, amplitude of low frequency fluctuation (ALFF), regional homogeneity (ReHo), and functional connectivity (FC) metrics as feature inputs for ML models. After conducting feature selection using a two-sample t-test, we established various ML classification models, including Extreme Gradient Boosting (XGBoost), Logistic Regression (LR), Random Forest (RF), Support Vector Machine (SVM), and Gaussian Naive Bayes (GNB). Comparative analysis of the model results revealed that the LR classifier based on BOLD signals could accurately distinguish flight cadets from the general population, achieving an AUC of 83.75% and an accuracy of 0.93. Furthermore, an analysis of the features contributing significantly to the ML classification models indicated that these features were predominantly located in brain regions associated with auditory-visual processing, motor function, emotional regulation, and cognition, primarily within the Default Mode Network (DMN), Visual Network (VN), and SomatoMotor Network (SMN). These findings suggest that flight-trained cadets may exhibit enhanced functional dynamics and cognitive flexibility.

## Introduction

The rapid expansion of China's civil aviation industry has heightened the significance of proficient pilot training. Due to the unique demands of the profession, every flight cadet must undergo systematic flight training before becoming a qualified pilot. This training process requires significant time, resources, and manpower, and it is

**Data availability statement:** Code and raw data related to this study can be found in the following public repositories: https://doi.org/10.6084/m9.figshare.28608278.v1

**Funding:** The study received funding from the Sichuan Science and Technology Program (Grant No. 2023NSFSC1183 to DY) and the Fundamental Research Funds for the Central Universities (Grant No. J2023-004 to DY).

**Competing interests:** The authors have declared that no competing interests exist.

invariably accompanied by assessment and elimination. Once a cadet is eliminated, the initial investments are lost, and the psychological impact on the eliminated cadets is considerable. Currently, there are relatively few reference indicators for the selection and training of flight cadets. The traditional approach involves a comprehensive judgment based on physical fitness, certain aspects of psychological endurance, and some flight performance metrics. However, these methods are limited by their lengthy duration, subjective evaluations, and lack of a robust theoretical foundation to determine suitability for subsequent flight training. Therefore, it is essential to incorporate more objective and efficient evaluation indicators in both pilot selection and cultivation processes while establishing a more effective assessment system.

In recent years, driven by the rapid advancements in brain science, researchers have employed resting-state functional magnetic resonance imaging (fMRI) to uncover that the brains of pilots experience both functional and structural alterations in comparison to those of the general population [1,2].This technique reflects the brain's activity state through blood oxygen level-dependent (BOLD) signal changes [3]. In brain network research, multiple networks are typically divided based on functional connectivity and anatomical structures, such as the Salience Network (SN), Default Mode Network (DMN), Central Executive Network (CEN), etc. Each network is responsible for specific cognitive and behavioral functions. The SN is mainly responsible for identifying and integrating sensory information, guiding attention and behavior. The DMN is most active when the brain is in a resting state and is involved in internal thought processes such as self – reflection, memory recall, and future planning. The CEN is associated with cognitive control and attention and is responsible for performing complex tasks, such as working memory, decision – making, and problem – solving. Previous studies have found that in terms of these brain networks, pilots exhibit enhanced functional connectivity between the SN, DMN, and CEN, while at the same time, the connectivity within each of these three networks decreases, and the temporal properties of functional dynamics increase [4,5]. A prior study identified notable elevations in spontaneous activity within brain regions linked to visual information processing and coordinated body movements in pilots by analyzing the Amplitude of Low-Frequency Fluctuation (ALFF) signals during the resting state [6]. Another study on degree centrality found that flight cadets, after training, exhibited increased degree centrality values in the left lingual gyrus and left middle frontal gyrus, which correlated with improvements in their executive abilities [7]. However, previous studies have primarily used univariate pattern analysis methods, which do not adequately capture the numerous detailed feature changes in resting-state fMRI data.

Recently, machine learning (ML) algorithms have gained prominence in the analysis of neuroimaging data due to their flexible modeling capabilities and their ability to identify complex relationships between input features and output data. ML can efficiently extract features from raw image data, offering significant advantages over traditional univariate pattern analysis. By using ML for data processing and analysis at an individual level, It becomes increasingly sensitive to minor alterations in fMRI data.[8,9], Leveraging this characteristic, some studies on resting-state brain function

have utilized ML algorithm models to compare and analyze resting-state fMRI data from different populations. This data-driven approach enhances our understanding of advanced cognitive neural mechanisms and intrinsic brain functions across diverse groups [10]. Consequently, ML algorithms have been introduced into various medical fields, demonstrating increasing clinical utility and relevance [11].

Currently, many studies on resting-state brain function focus on patient populations, such as those with Alzheimer's disease or mental illness [12,13]. This is because the physiological structure and morphology of these patients' brains have undergone significant changes compared to healthy individuals, and ML methods can efficiently capture areas of brain pathology for clinical diagnosis [14]. However, there is still a lack of research using ML methods to study the brain function of healthy populations such as pilots. In this study, all the research subjects were from the healthy population. However, the flight training that flight cadets participate in encompasses elements such as high – pressure decision – making, multi – sensory integration (including aspects like vision, hearing, and spatial perception), and rapid cognitive switching, which are significantly different from the theoretical or operational tasks that ground cadets engage in. For example, in the aircraft cockpit, flight cadets must process a vast amount of information in real – time, such as instrument panel data, tower signals, and radio communication content, and make quick decisions. This intense cognitive training is highly likely to have a unique impact on brain functions. Therefore, exploring the changes in the brain functions of healthy flight cadets can lay the foundation for a deeper understanding of the mechanisms by which this special vocational training shapes the brain, and also contribute to the development of more efficient training strategies. Given that the changes in brain functions among healthy individuals are often more subtle and difficult to detect, this study employs the ML method to conduct an in – depth investigation of the fMRI data of flight cadets' brains. Therefore, this study extracted mainstream indicators of resting-state fMRI, including ALFF, functional connectivity (FC), and Regional Homogeneity (ReHo) as feature inputs for an ML model [15,16]. Additionally, voxel-level BOLD signals were extracted from the collected raw resting-state fMRI data as parameter indicators. The BOLD signal, as the raw voxel signal, is essentially a time series signal. Changes in blood flow alter the ratio of oxygenated to deoxygenated hemoglobin in brain regions, affecting the uniformity of local magnetic fields and generating changes in the BOLD signal. This indirectly reflects the activation of high-activity brain areas in fMRI results [17].

Therefore, this study aims to use various resting-state fMRI parameter indicators and machine learning methods to accurately distinguish flight cadets from healthy individuals without flight training at an individual level. This study is of an exploratory nature. Integrating previous relevant research, we hypothesize that flight training will bring about changes and exert impacts on the brain's cognitive abilities. Moreover, we assume that machine – learning methods can capture the differences in various brain fMRI indices between flight cadets and ground cadets, thus enabling accurate classification. Additionally, we aim to analyze features that significantly contribute to classification decisions from a neurobiological perspective by combining resting-state fMRI indicators to explore intrinsic changes in brain function after systematic flight training. Furthermore, we established classification models using five different machine learning algorithms and conducted a horizontal comparison of four extracted resting-state fMRI indicators. To our knowledge, this is the first exploration of changes in brain function based on resting-state fMRI data from flight cadets using machine learning.

## Materials and methods

### Participants

The participants were selected from the Civil Aviation Flight University of China(CAFUC). The recruitment announcement detailed the experiment's content, screening criteria, and potential hazards and risks. A total of 79 cadets participated in the MRI scanning: 40 flight cadets, all of whom had completed at least 235 hours of flight training at the CAFUC, obtained private and commercial pilot licenses issued by the Civil Aviation Administration of China, and were qualified to independently fly small airplanes. The remaining 39 subjects formed a control group consisting of trainees from the ground program. In this study, both the experimental group and the control group were composed of cadets from the same

enrollment batch. These cadets had similar entrance examination scores. Throughout their undergraduate studies, the learning content, total credit hours, and theoretical class hours of the two groups of cadets were essentially equivalent. In the initial selection of flight cadets, the emphasis was on physical fitness and visual acuity, and the selection requirements in other aspects were the same as those for ground cadets. The control group consisted of cadets who received no flight training and only participated in ground-based theoretical courses (such as aviation theory, meteorology, and navigation principles). All ground-based cadets came from the engineering discipline of the same institution, and possessed neither a flight license nor any experience in flight simulation. Their training content significantly differed from that of the flight cadets, focusing primarily on theoretical learning and ground operation skills, rather than practical flight tasks. The recruitment period for participants commenced in June 2022 and concluded in November 2022. The acquisition of magnetic resonance imaging data began in June 2022 and ended in November 2022.

Inclusion criteria for all subjects were: (1) no metal objects in the body; (2) in good health at the time of participation; (3) all male; (4) aged 21−25 years; (5) right-handed; (6) no history of psychiatric disorders; (7) no drug or alcohol dependence; (8) no history of traumatic brain injuries, cerebrovascular disease, or chronic pain; (9) all holding undergraduate degrees in engineering; (10) All participants provided informed consent and voluntarily signed a consent form before the experiment commenced. The research adhered to the principles of the Declaration of Helsinki. The Ethics Committee of the University of Electronic Science and Technology of China (Chengdu, China) approved the experimental procedures under No. 1420200408−07.

## Data acquisition and preprocessing

All data in this study were collected using a 3-Tesla MRI scanner (DISCOVERY MR 750, GE) at the Magnetic Resonance Imaging Center of the University of Electronic Science and Technology of China.

The T1-spoiled gradient recalled echo pulse sequence was employed to obtain high-spatial-resolution structural images. The scan parameters included a repetition time (TR) of 5.952 ms, an echo time (TE) of 1.964 ms, a flip angle of 9°, a matrix size of 256 × 256, a slice thickness of 4 mm with no gap, a field of view (FOV) of 25.6 cm × 25.6 cm, and a total of 154 slices were acquired, the voxel size = 1 mm × 1 mm × 1 mm.

The functional images at rest were obtained using a gradient-echo echo-planar imaging sequence. Subjects were instructed to lie quietly with their eyes closed for the duration of the 510-second scan while remaining awake. The scan parameters were: TR = 2000 ms, TE = 30 ms, flip angle = 90°, matrix size = 64 × 64, slice thickness = 4 mm with a 0.4 mm gap, FOV = 24 cm × 24 cm, 35 slices per volume, and a total of 255 volumes acquired, the voxel size = 3 mm × 3 mm × 3 mm.

The RestPlus toolbox on the MATLAB 2022b platform was utilized for preprocessing resting-state fMRI data. The preprocessing steps for resting-state fMRI data are as follows: (1) removal of the first five time points to minimize interference from unstable magnetization vectors; (2) temporal slice correction to mitigate scan time differences between slices; (3) head motion correction by excluding samples with translation >2 mm or rotation >2°; (4) The structural and functional images of the subjects were aligned, and the structural images were segmented to obtain white matter, gray matter, and cerebrospinal fluid; (5) regression of covariates, including average cerebrospinal fluid signal, average white matter signal, and head motion signals derived from the Friston-24 parameter model; (6) Realigning individual brain images to the standard space of the Montreal Neurological Institute (MNI) at a resolution of 3 mm × 3 mm × 3 mm through spatial normalization; (7) spatial smoothing using a Gaussian filter to reduce spatial noise and remove signals outside the frequency range of 0.01–0.08 Hz; and (8) removal of linear drift.

## Feature extraction

The RestPlus 1.28 toolkit, SPM12, and custom MATLAB 2022b code were used to compute the ALFF, ReHo, FC, and BOLD signal metrics in this study. The BOLD signals and FC metrics were obtained from 116 brain regions (Table 1, the 90 cerebral and 26 cerebellar regions) based on the automated anatomical labeling (AAL) [18]. The BOLD signals of

**Table 1. Full names of all brain regions in the 116 Regions Defined in Automated Anatomical Labeling Template and their abbreviations.**

| Region Name | Abbreviation | Region Name | Abbreviation |
|---|---|---|---|
| Precentral gyrus | PreCG | Supramarginal gyrus | SMG |
| Superior frontal gyrus, dorsolateral | SFGdor | Angular gyrus | ANG |
| Orbitofrontal cortex (superior) | ORBsup | Precuneus | PCUN |
| Middle frontal gyrus | MFG | Paracentral lobule | PCL |
| Orbitofrontal cortex (middle) | ORBmid | Caudate | CAU |
| Inferior frontal gyrus (opercular) | IFGoperc | Putamen | PUT |
| Inferior frontal gyrus (triangular) | IFGtriang | Pallidum | PAL |
| Inferior frontal gyrus (orbital) | ORBinf | Thalamus | THA |
| Rolandic operculum | ROL | Heschl gyrus | HES |
| Supplementary motor area | SMA | Superior temporal gyrus | STG |
| Olfactory cortex | OLF | Temporal pole (superior) | TPOsup |
| Superior frontal gyrus (medial) | SFGmed | Middle temporal gyrus | MTG |
| Orbitofrontal gyrus (medial) | ORBmed | Temporal pole (middle) | TPOmid |
| Rectus gyrus | REC | Inferior temporal gyrus | ITG |
| Insula | INS | Cerebelum_Crus1 | CRBLCrus1 |
| Anterior cingulate gyrus | ACG | Cerebelum_Crus2 | CRBLCrus2 |
| Middle cingulate gyrus | DCG | Cerebelum_3 | CRBL3 |
| Posterior cingulate gyrus | PCG | Cerebelum_4_5 | CRBL45 |
| Hippocampus | HIP | Cerebelum_6 | CRBL6 |
| Parahippocampal gyrus | PHG | Cerebelum_7b | CRBL7b |
| Amygdala | AMYG | Cerebelum_8 | CRBL8 |
| Calcarine cortex | CAL | Cerebelum_9 | CRBL9 |
| Cuneus | CUN | Cerebelum_10 | CRBL10 |
| Lingual gyrus | LING | Vermis_1_2 | Vermis12 |
| Superior occipital gyrus | SOG | Vermis_3 | Vermis3 |
| Middle occipital gyrus | MOG | Vermis_4_5 | Vermis45 |
| Inferior occipital gyrus | IOG | Vermis_6 | Vermis6 |
| Fusiform gyrus | FFG | Vermis_7 | Vermis7 |
| Postcentral gyrus | PoCG | Vermis_8 | Vermis8 |
| Superior parietal gyrus | SPG | Vermis_9 | Vermis9 |
| Inferior parietal lobule | IPL | Vermis_10 | Vermis10 |

116 brain regions were obtained from the preprocessed resting-state fMRI image data, which contained 250 time points, forming a BOLD time-series signal for each subject. FC was defined as the correlation between the mean BOLD time series of different brain regions, measured by Pearson's correlation coefficient. FC can reflect the degree of connection tightness between various brain regions in the brain functional network, which helps to understand how different regions of the brain cooperate with each other functionally. To facilitate further analysis and statistics, the correlation coefficients were converted to z-values using Fisher's r-to-z transformation. A 116 × 116 symmetric matrix of functional connectivity was calculated from the preprocessed images for each subject, and feature selection was performed on the upper triangular elements of this matrix, resulting in a total of (116*115)/2 = 6670 features for functional connectivity. In this study, the calculation of ALFF involves performing a Fourier transform on the time series of each voxel to obtain the power spectrum within the low-frequency range (0.01–0.08 Hz). The square root of this power spectrum is then computed to derive the ALFF value for each voxel. This value effectively reflects the intensity of spontaneous neural activity in the brain and provides a critical metric for assessing the brain's intrinsic functional state. ReHo, on the other hand, is computed using

Kendall's coefficient of concordance (KCC). Specifically, the KCC value is calculated by comparing the time series of each voxel with those of its neighboring voxels, which measures the temporal coherence within local brain regions. This metric is crucial for understanding the synchrony of neural activity within localized brain areas, contributing to the exploration of the characteristics of local brain functional integration. ALFF and ReHo metrics were calculated using the RestPlus toolkit, yielding 116 features each per subject. The BOLD signal was calculated using MATLAB code based on SPM-related expansion functions, resulting in $250 \times 116 = 29,000$ features per subject.

## Feature selection

Feature selection in this study involved the use of a two-sample t-test. The two-sample t-test is a commonly used algorithm in fMRI research and has demonstrated good results in feature screening for various neuroimaging studies utilizing ML algorithms [19]. In feature selection, the two-sample t-test assesses the extent to which each feature differs between different categories (e.g., positive and negative categories).For each extracted feature, we use a two-sample t-test to compare the mean differences between the positive and negative category samples and assesses the statistical significance of these differences by calculating the resulting *p*-values. The issue of multiple comparisons is a primary challenge faced by t-tests in high-dimensional data analysis. Each individual t-test introduces a probability of error, and when conducting numerous tests, the cumulative false positive rate (Type I error) increases, potentially resulting in a large number of false significant features. To address this, we introduced Levene's Test prior to conducting the t-tests, primarily to assess the homogeneity of variances between two groups. The t-test assumes that the variances of the two sample groups are equal (i.e., homogeneity of variance). Levene's Test evaluates whether the variance is the same across groups by calculating the deviation of each data point from its group mean. If the result of Levene's Test is not significant ($p > 0.05$), the assumption of homoscedasticity is upheld, allowing for the use of the standard t-test. However, if the result is significant ($p \leq 0.05$), the assumption is violated, indicating unequal variances, and in this case, the Welch t-test is required. The Welch t-test, which does not assume equal variances, is more robust in cases of heteroscedasticity, preventing erroneous conclusions that might arise from unequal variances. The Scipy software package (version 1.11.1, https://scipy.org/) was used to implement the feature screening algorithm for the two-sample t-test. Custom code was written to execute this feature selection process.

## ML model establishment

Five algorithms were employed to develop and compare prediction models. For ALFF, ReHo, FC, and BOLD metrics, different potential predictor variables were selected as feature inputs to the ML model based on a two-sample t-test feature selection approach. The methods used to construct the classification models included Gaussian Naive Bayes (GNB), Support Vector Machine (SVM), Random Forest (RF), Logistic Regression (LR), and Extreme Gradient Boosting (XGBoost). The five categories of models we chose are widely utilized in fMRI classification tasks and are characterized by high accuracy and robustness [20–23]. For example, RF, SVM, and XGBoost perform outstandingly when handling complex data [14,19,24,25]. LR, being a classic linear classification model, is simple, easily comprehensible, and highly interpretable. GNB, based on Bayes' theorem, features high computational efficiency and can rapidly process large – scale data, especially in data scenarios where features are relatively independent [26,27]. All prediction models were implemented in Python 3.9. XGBoost was implemented using the "xgboost 1.2.1" package, while the other models were implemented using the "sklearn 0.22.1" package. The optimal hyperparameters of the machine learning model are determined using the grid search method, with the training and test sets divided in a ratio of 4:1.

## Model evaluation and validation

The model's predictive ability was evaluated using the standard Receiver Operating Characteristic (ROC) curve, area under the curve (AUC), accuracy, sensitivity, specificity, and confusion matrix. Before performing model training, 20% of the data was randomly selected as the test set, while the remaining 80% was used as the training set. The training set

was then randomly divided into five groups. In each iteration of cross-validation, four groups were selected for training and the remaining one was used as the validation set. The model that performs best on the validation set was then deployed to the test set, and the results were quantified using mean ROC curves. The prediction accuracy was assessed using the obtained AUC and confusion matrix. Furthermore, a permutation test was performed to assess the significance of the model's predictions.

## Results

### Participant demographics

Three subjects were excluded from the trial due to being left-handed and having excessive head movements. As a result, the study ultimately included data from 39 participants in the flight group and 37 participants in the control group. Table 2 presents the demographic data of the subjects who were finally included in the study.

### The quantity of features

In this study, four kinds of resting-state fMRI indicators were extracted, and when two-sample t-tests were conducted for different indicators, setting the appropriate thresholds was crucial to the effect of feature screening. The $p$-value is a crucial metric used to assess the significance of differences between groups for a given feature. In selecting the $p$-value threshold, we adhered to commonly accepted significance levels, such as 0.05, a widely recognized standard for distinguishing statistically significant from non-significant features. While a $p$-value of 0.05 has been the conventional threshold in previous research, with the advent of machine learning and high-dimensional datasets, many recent studies advocate for stricter thresholds. For instance, Madsen et al. [13] argue that when analyzing neuroimaging data, lower p-value thresholds can more effectively reduce false positives arising from multiple comparison issues. Our study involves high-dimensional data, where the number of features exceeds the sample size. In such cases, using a higher $p$-value threshold (e.g., 0.05) may result in a considerable number of false positives. Therefore, to mitigate this risk, we adopted stricter thresholds (e.g., 0.01 or 0.005), adjusted according to the volume of data across different metrics. Table 3 shows the threshold levels of different resting-state fMRI indicators and the number of features after screening.

**Table 2. Comparative analysis of demographic information among the two groups.**

| Characteristics | Flight Cadets (N = 39) | Controls (N = 37) | *T*-value | *P*-value |
|---|---|---|---|---|
| Age (years) | 22.9 ± 0.96 | 22.4 ± 0.95 | 1.385 | 0.065 |
| Gender (% male) | 100 | 100 | — | — |
| Education (years) | 16 | 16 | — | — |
| Handedness(% right) | 100 | 100 | — | — |
| Total flight time (hours) | 240.56 ± 8.74 | 0 | — | — |

**Table 3. Here is the breakdown of the number of features obtained from the four resting-state fMRI metrics (BOLD, ALFF, ReHo, and FC) at various thresholds of the two-sample t-test.**

| | FC | BOLD | ALFF | ReHo |
|---|---|---|---|---|
| $p$ | 0.005 | 0.01 | 0.05 | 0.05 |
| Number of features | 141 | 40 | 17 | 12 |

**Notes:** In this study, features are defined as quantitative metrics extracted from resting-state fMRI data, including BOLD signals, FC, ALFF, and ReHo.

## Comparison of multiple classification models

In this section, the present study evaluated the performance of various resting-state fMRI indicators on different classifiers and compared their performance across different indicators. After conducting a two-sample t-test for feature selection on four types of resting-state fMRI indicators, an ML prediction model was established based on the obtained optimal feature subset. Among them, GNB and LR classifiers demonstrated better classification performance than SVM, RF, and XGBoost classifiers under the same type of resting-state fMRI indicator. When considering various types of resting-state fMRI indicators, the ML model with BOLD signals as input features achieved the best classification performance. FC performed worse than BOLD signals in the ML model, while ALFF and ReHo signals as input features resulted in relatively poor classification performance in the ML model.

As shown in Fig. 1, when comparing ROC curves, the AUC of BOLD signal indicators are all larger than those of FC, ALFF, and ReHo. Among the five ML models using BOLD metrics as feature inputs, their classification effects are all significantly higher than the results of random classification, with LR achieving the highest accuracy and AUC of 83.75% and 0.93, respectively. FC's overall effect on the classification of the five ML models is higher than that of random classification, with GNB having the best accuracy and AUC of 73.75% and 0.82. The performance of ALFF and ReHo metrics on ML classification was close to that of the randomized classification results.

By calculating true positive examples (TP), false positive examples (FP), true negative examples (TN) and false negative examples (FN), we obtained the confusion matrix of ML models, visually quantifying the performance of the classification models. As shown in Fig 2, We generated the confusion matrix for the machine learning models with the best performance results for each indicator and found that the overall classification effect is still best for the BOLD signal indicator.

In terms of sensitivity, the ML model based on BOLD signal metrics performs best, with the GNB classifier achieving the highest sensitivity of 0.85. For specificity, the LR classifier based on BOLD signal metrics performs the best, reaching 0.82. For these two evaluation metrics, FC performs generally well, while ALFF and ReHo are generally low. We make a visual comparison in Fig 3.

## Feature weights and key brain regions

Through permutation testing, this study finds that the BOLD signal and FC metrics exhibit statistically significant classification performance in ML models, whereas the ALFF and ReHo metrics do not. Consequently, subsequent analyses will focus exclusively on the BOLD signal and FC metrics. Preliminary feature selection yielded 40 and 141 features for the BOLD signal and FC metrics, respectively. To better investigate the contribution of these features to ML classification, this study introduces the feature weight parameter from the SVM model, with feature weights positively correlated with classification contribution. Utilizing five-fold cross-validation, each feature weight is calculated by summing the absolute values of the weights from the cross-validation and then averaging them. By tracking sequence indices, it was determined that the 40 BOLD signal features originate from 21 brain regions, and the 141 FC features correspond to 56 brain regions. Fig 5(a) displays the BOLD signal weights for ML classification across the 21 brain regions, while Fig 5(b) presents the FC weights. Due to the large number of FC features, only those with weights exceeding a standard threshold (i.e., the mean weight plus the standard deviation) are presented [14]. Key features with weights above this threshold are deemed crucial for ML classification, covering essential brain regions and connections. As shown in Fig 5, nine key brain regions and twelve critical connections were identified. The key brain regions include the Right Orbitofrontal Cortex superior (RORBsup), Left Middle Occipital Gyrus (LMOG), Left Fusiform Gyrus (LFFG), Right Postcentral Gyrus (RPoCG), Left Angular Gyrus (LANG), Left Precuneus (LPCUN), Left Inferior Temporal Gyrus (LITG), Left Temporal Pole middle (LTPOmid), and the Right Cerebellum Crus2 (RCRBLCrus2).

## Brain network segmentation

The human brain's various regions are not isolated and unconnected; rather, they constitute a large-scale network of interrelated structures. Adopting a network-based approach is instrumental in understanding the organization of brain

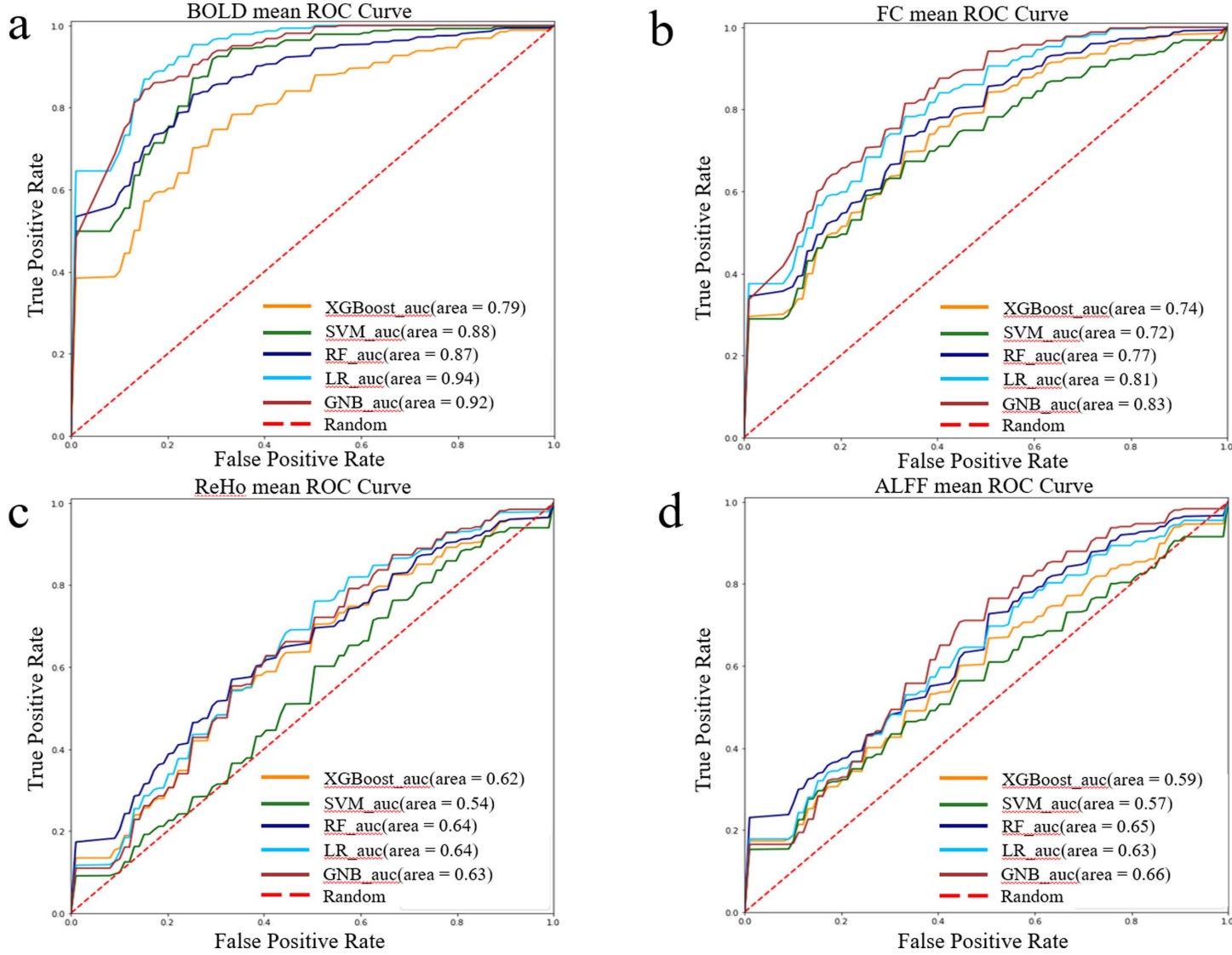

**Fig 1. ROC comparison curves for five distinct ML models.** Distinct colors symbolize varied ML models, with the central dashed line denoting random classification. The area under the ROC curve and bounded by the axes is commonly referred to as AUC.(a) The ROC curve and AUC comparison of various ML models for the BOLD indicator.(b) The ROC curve and AUC comparison of various ML models for the FC indicator.(c) The ROC curve and AUC comparison of various ML models for the ReHo indicator. (d) The ROC curve and AUC comparison of various ML models for the ALFF indicator.

functions [28]. Prior research has proposed numerous intrinsic brain network models. To better analyze the brain regions critical in classification, this study utilizes network templates [29], combined with the brain region network partitioning method by Sun et al. (K-Means clustering) [14], categorizing AAL brain regions with significant contributions to classification results into seven brain networks. As illustrated in Fig 6(a), when BOLD signals are used as ML input features, the frontoparietal network (FPN) and default mode network (DMN) predominantly contribute, followed by notable contributions from the visual network (VN) and somatomotor network (SMN). Contributions from the cerebellum (Cer) are minimal, with no features from the dorsal attention network (DAN) and limbic network (LN). Key brain regions are primarily located in the DMN and VN, with a higher proportion. As depicted in Fig 6(b), when FC is used as the ML input feature, connections within the VN, DMN, and SMN dominate, with substantial contributions

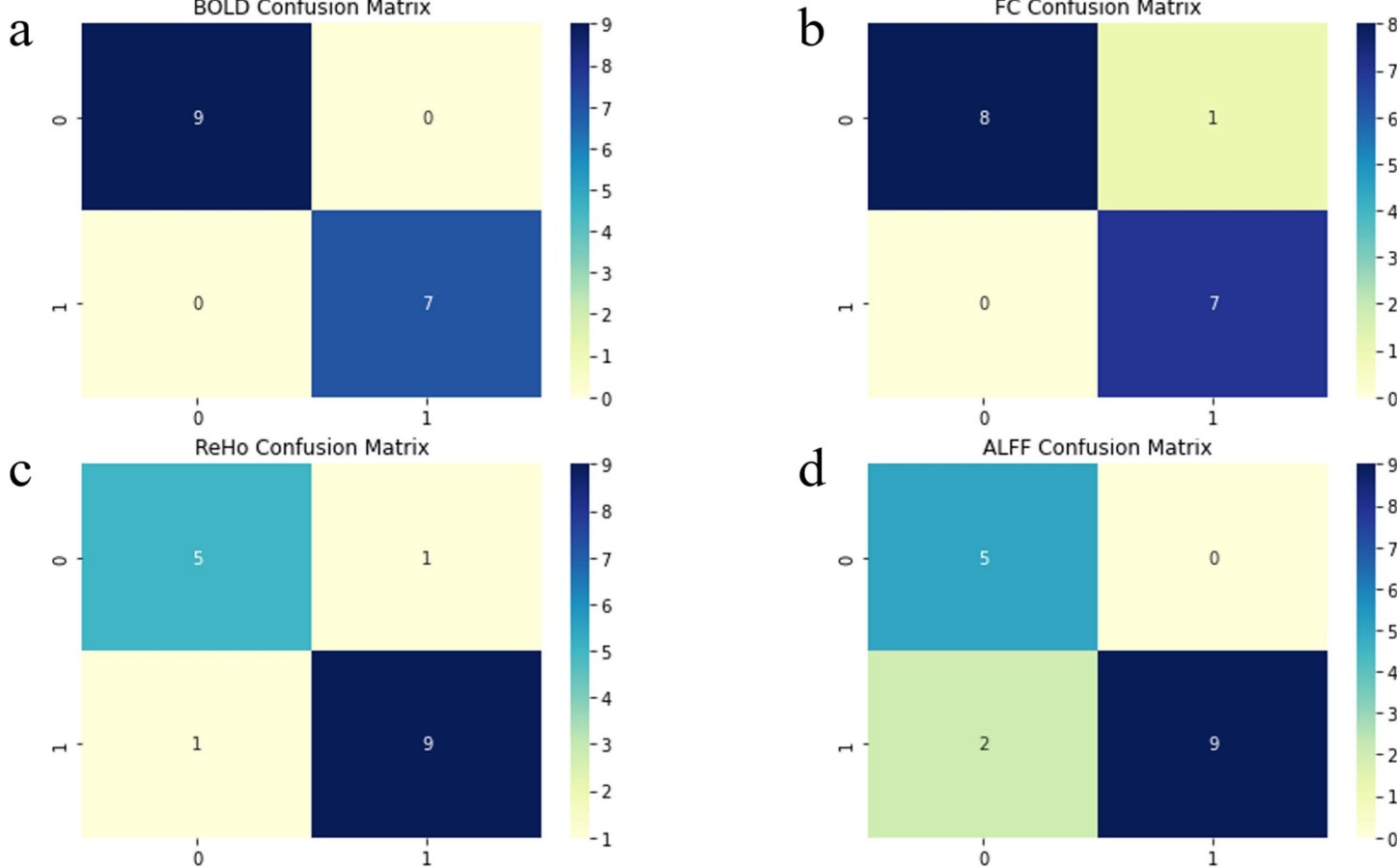

**Fig 2. The confusion matrix of the best performing model is plotted against different resting-state fMRI metrics.** (a) The confusion matrix for the LR model utilizing the BOLD metric.(b) The confusion matrix for the GNB utilizing the FC metric.(c) The confusion matrix for the RF utilizing the ReHo metric. (d) The confusion matrix for the GNB utilizing the ALFF metric.

from the FPN and cerebellum. Functional connectivity features associated with the DAN and LN are comparatively minimal.

## Discussion

In this research, five different machine learning classification models were developed for each of the four unique resting-state fMRI indicators. The comparative analysis of these resting-state fMRI metrics demonstrated that the BOLD signal, which indicates the activation level of brain regions, outperformed the FC, ALFF, and ReHo metrics as a classification feature. Conversely, the vertical comparison of similar metrics across the five machine learning classifiers revealed that the LR, which is apt for linear relationships among features, and the GNB, which presumes feature independence, yielded superior results for this dataset. In contrast, the SVM, XGBoost, and RF, which are adept at addressing non-linear relationships and handling high-dimensional, large-scale data, performed relatively poorly.

The focus of this study is on flight cadets. The flight training of flight cadets encompasses not only the enhancement of physical fitness and piloting skills but also the development of cognitive abilities in diverse and complex environments, such as decision-making, hypothesis formation, and reasoning during flight. Previous research has demonstrated that cognitive training can influence brain function factors and, to some extent, alter brain morphology [30,31]. Studies have

**Fig 3. Various ML evaluation indicators of four indicators BOLD signal, FC, ALFF, ReHo on five ML models (XGBoost, SVM, RF, GNB, LR).** Given the limited sample size and the specificity of the study population, this research employs permutation tests to assess whether the model's performance is attributable to chance. By randomly shuffling the dataset labels, if the model's performance significantly deteriorates on the permuted data, it suggests that the model has effectively learned meaningful patterns from the data. The distribution of the results and *p*-values from the permutation test for the optimal model across the four resting-state fMRI metrics are presented in Fig 4. The permutation test (repetition number: 10,000) revealed that the actual classification accuracy of the GNB model, based on BOLD signal and FC as input features, was superior to that of the randomly labeled data ($p < 0.05$). Conversely, the RF model, employing ALFF and ReHo as feature inputs, failed to pass the permutation test ($p > 0.05$), indicating that its classification results were statistically insignificant, rendering its performance unreliable.

indicated that individuals with specialized skills training exhibit significant changes in brain morphology and regional brain function compared to the general population. Bermudez and colleagues discovered that musicians, compared to ordinary individuals, have a notable increase in gray matter in the right superior temporal gyrus and the lateral surface of the transverse temporal gyrus [32]. Huang et al. discovered that there was a significant increase in gray matter density and noticeable changes in various brain regions among gymnastics world champions compared to the normal control group [33]. Flight training represents a form of specialized skill training, where repetitive and unexpected movements continually stimulate brain regions associated with emotional control, cognitive function, operational response, and other related areas. Causse et al. observed that high uncertainty during flight task decisions greatly increases the activity of certain brain regions [34], Thus, As a result of systematic flight training, the brains of flight cadets undergo structural and functional changes, making brain areas responsible for emotional regulation and cognitive function more active than those

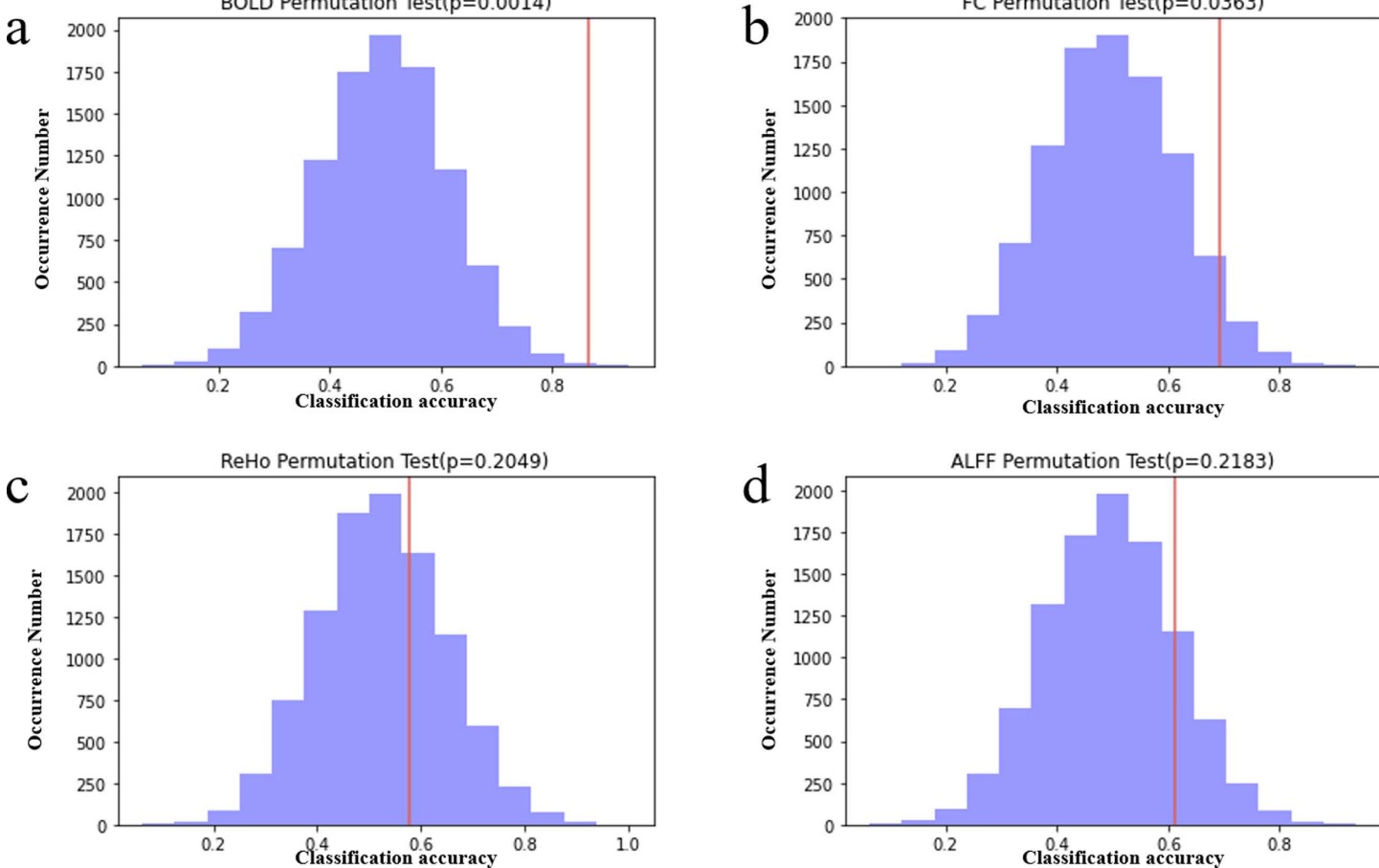

**Fig 4. BOLD signal, FC, ALFF, and ReHo metrics based on optimal model permutation test results.** The red dashed line indicates the accuracy of the model using real labels.(a) Permutation tests based on the BOLD metric.(b) Permutation tests based on the FC metric. (c) Permutation tests based on the ReHo metric. (d) Permutation tests based on the ALFF metric.

in the ground control group without flight training. These altered brain regions can be identified through the analysis of resting-state fMRI indicators [6,7].

In this study, utilizing the BOLD signal and FC index, the ML method was employed to determine the regional contribution weights, identifying 9 key brain regions and 12 key connections. The combination of these indicators revealed that most of these critical brain regions and connections are part of the DMN, SMN, and VN, with the DMN regions being the most prevalent. This indicates that the activation and functional connectivity of brain regions within the DMN in flight cadets differ from those in ordinary individuals. This finding is consistent with CHEN et al.'s study, which found changes in the internal functional connectivity of the DMN in pilots [5].

The DMN is a network of brain regions active during rest, involved in functions such as self-reflection, mind-wandering, recalling personal experiences, and future planning. It is characterized by continuous high activation during rest and inhibition during task performance. The DMN plays a crucial role in introspective activities, including self-reflection, recall, imagination, future event planning, and social cognition [35]. This study identified the locations and brain networks of key brain regions that facilitate ML in distinguishing between flight cadets and ordinary individuals, as shown in Fig 7.

The DMN regions include the angular gyrus (ANG), precuneus (PCUN), and middle temporal gyrus (TPOmid). The ANG is involved in processing visuospatial information, such as understanding spatial orientation and navigation, aiding

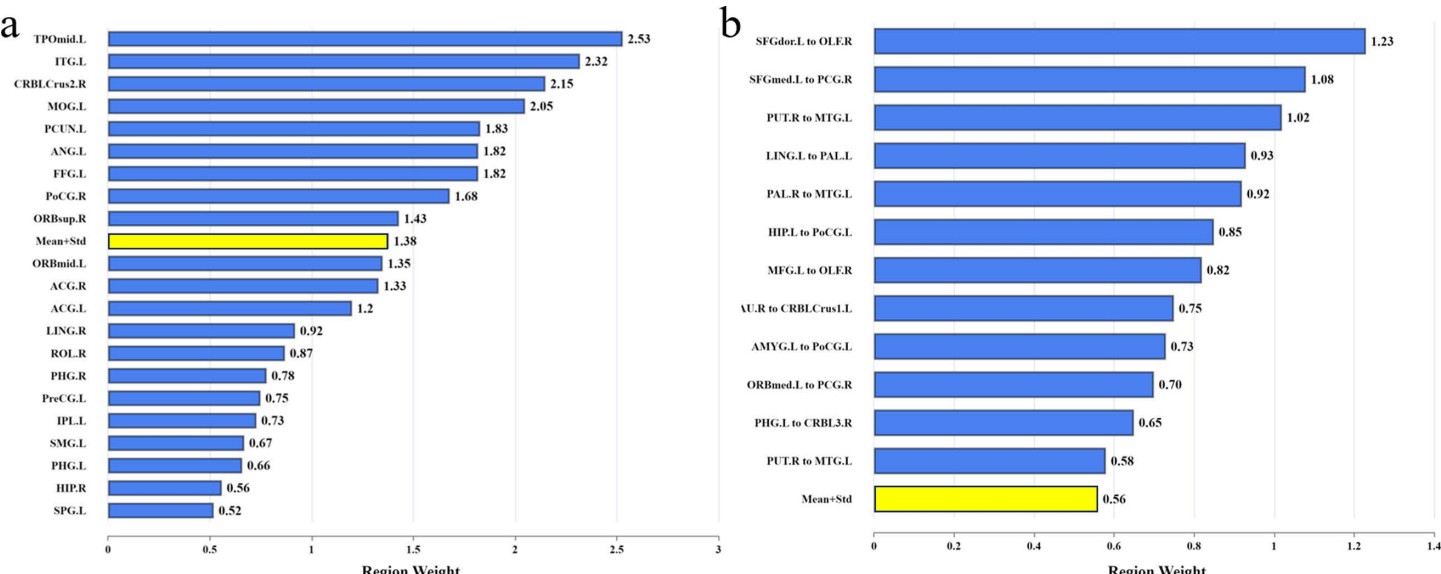

**Fig 5. High weight features obtained by SVM model. The yellow bar serves as a baseline for screening key brain regions and key connections.** (a) The brain regions of high contribution features obtained based on the BOLD metric. (b) The brain regions of high contribution features obtained based on the FC metric.

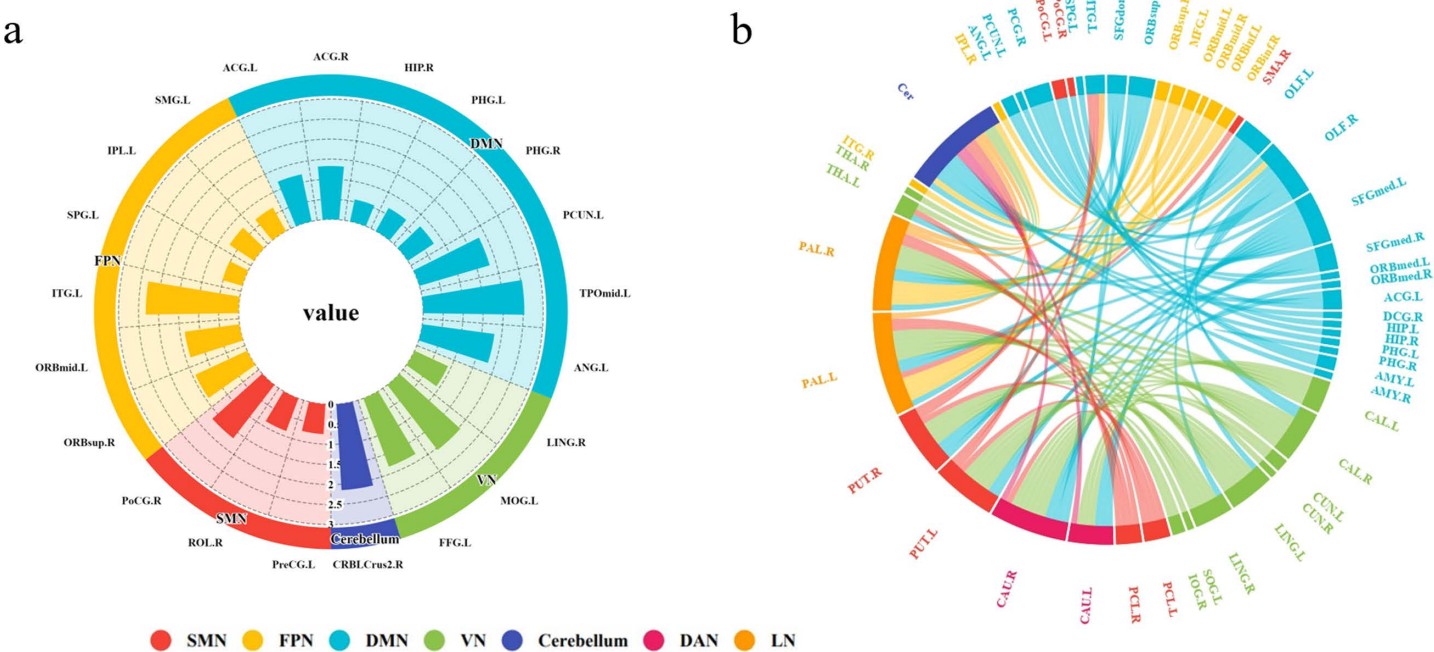

**Fig 6. The features derived from a two-sample t-test, along with their corresponding brain regions and brain networks.** Different colors represent different brain networks. (a) The features obtained from the screening based on the BOLD metrics, as well as the brain regions and brain networks to which these features belong and their weights. (b) The features obtained from the screening based on the FC metrics, as well as the brain regions and brain networks to which these features belong and the connections between them.

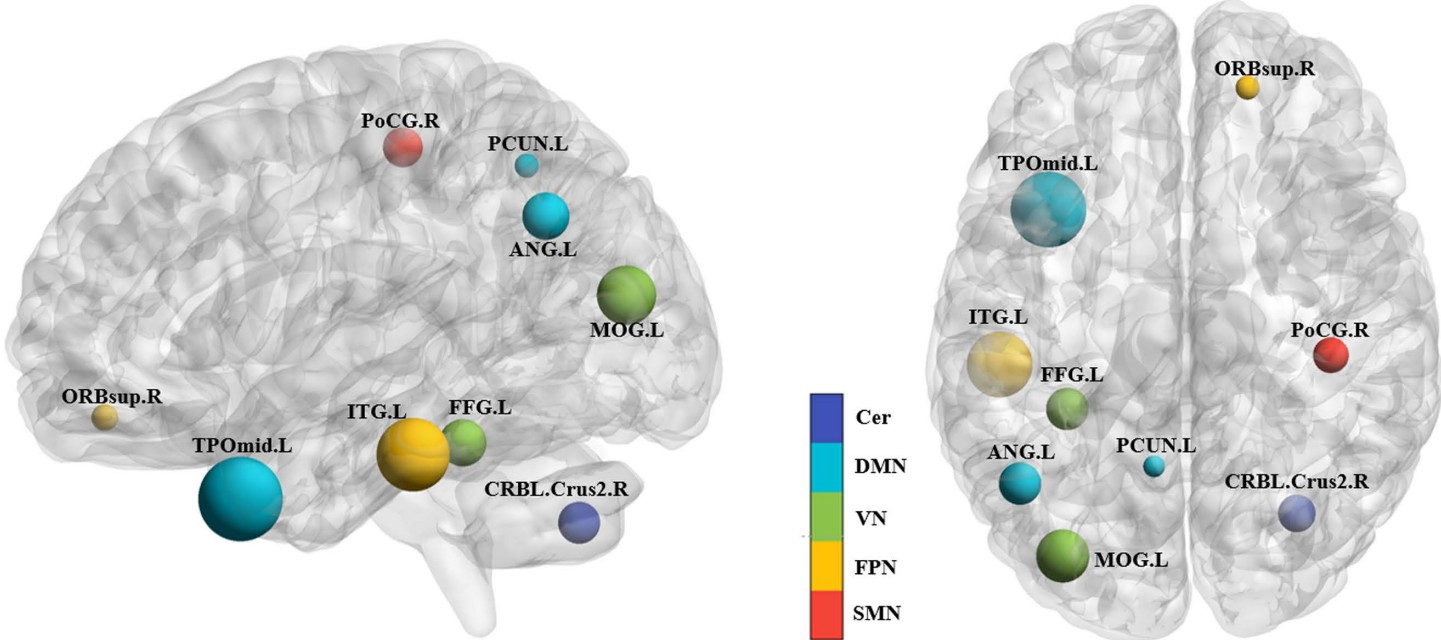

**Fig 7. Spatial distribution and associated brain networks of key regions in BOLD.** The various colors correspond to distinct cerebral networks. The size of the sphere represents the size of the contribution weight in the classification.

individuals in comprehending the position and orientation of objects in space. It also integrates cross-sensory information, combining visual data with auditory or tactile information to form a comprehensive environmental understanding, crucial for pilots in enclosed flying environments [36]. PCUN is a region in the medial parietal lobe of the brain, associated with various complex cognitive functions related to introspection and self-related thinking. The precuneus plays a central role in the DMN [37], participating in the reflection on previous experiences and envisioning potential future situations [38]. Previous research has indicated that PCUN is involved in spatial awareness and navigation, aiding individuals in understanding and orienting themselves in the environment [39]. Additionally, PCUN has been associated with the interpretation of visual information, particularly in integrating visual information and understanding complex visual scenes. The TPOmid is located in the middle temporal gyrus, a crucial region for language processing, especially in vocabulary comprehension, semantic processing, and verbal memory. It is engaged in handling tasks related to language, such as combining lexical and semantic information, and the middle temporal gyrus also plays a role in auditory information processing, including understanding of speech and non-speech, and it is involved in the conversion of auditory signals into easily understandable linguistic content [40,41], and activation of auditory- and language-related brain regions may be a result of the need for pilots to communicate with ground towers at all times during piloting and the need to understand and decoding, which involves acoustic information processing and conversion.

The FPN is mainly linked to advanced cognitive processes like controlling attention, managing working memory, making decisions, and resolving problems. It is believed to coordinate activities between disparate brain regions to facilitate complex cognitive tasks, with its main constituent regions situated in the frontal and parietal areas [29]. The ORBsup is a part of the prefrontal lobe, plays a role in value-based decision making, which encompasses an individual's process of maximizing gains through appropriate actions or choices, and involves sensitivity to risks, rewards, and punishments, thereby influencing decision-making processes. Damage to this brain region may result in difficulties with impulse suppression and event comprehension. Additionally, ORBsup contributes to the regulation of emotional and social behavior

[42,43]. The ITG, within the FPN, is implicated in language comprehension, including the visual recognition of words and semantic processing of language [44], and also participates in decision making and emotion regulation. It involves higher cognitive functions such as the evaluation of visual and verbal information and the control of emotional responses [45].

The VN comprises a group of brain areas that are in charge of handling visual data within the brain.. These regions are interconnected and function through a complex neural network to ensure accurate and coordinated visual information processing. The VN's functions extend beyond visual perception to include higher cognitive functions such as attention, memory, and decision-making. The FFG participates in both the VN and DMN brain networks and is part of the visual association cortex. It combines primary visual information with higher-level visual processing, aiding in understanding the shape, size, and spatial relationships of objects [46,47]. The MOG, located in the occipital lobe after the central sulcus (Calcarine Sulcus) in the visual cortex, is a vital component of the VN. It has a critical function in the processing of visual data, especially shapes, colors, and complex visual scenes [48]. XU et al. discovered that the mean ALFF in the left MOG of pilots was higher than in the general population [6]. The MOG is also involved in spatial information processing [48], and significant brain activation is often detected during active driving [49]. This activation in pilots' MOG may relate to the continuous need to process extensive spatial and orientation information during flight.

The SMN is a brain network involved in sensory and motor information processing. It primarily converts sensory inputs into motor outputs, enabling individuals to respond appropriately to external stimuli. Multiple brain regions comprise the SMN, working in concert to support sensorimotor functions. The PoCG is a key region identified in this study, is an essential part of the SMN. The PoCG is essential for controlling motor functions [50] and is also associated with auditory signals [51]. The AAL116 template used in this study includes the cerebellum, located in the posterior part of the brain, which is primarily responsible for coordinating and fine-tuning movements. The cerebellum serves as a significant regulatory center for movement, with extensive afferent and efferent motor regulatory information [52]. Recent studies have shown that the cerebellum is involved not only in motor control but also in higher cognitive and affective functions, such as working memory, language, visuospatial processing, and executive functions [53,54].

As shown in Fig 8, the previously identified key connections are located in brain networks highly correlated with emotional control, audio-visual processing, cognitive switching, and spatial perception abilities, similar to the networks to which the key brain regions belong. We hypothesize that this is closely related to the daily flight training of flight cadets. At the Civil Aviation Flight College of China, cadets are constantly under high stress during flight training, and sudden emergency scenarios in simulation programs enhance their ability to handle emergencies and quickly adjust their emotions. During training, cadets need to capture various types of information in real-time, such as instrument readings, radio signals, tower communications, and ground landmarks. This information involves multiple senses, including sensation, hearing, and vision, initially processed by the VN and SMN and then further processed by the DMN and FPN [55]. Cadets must continuously perform cognitive transformations during flight to make correct operational decisions, a process intricately linked to the DMN [56,57]. They need to integrate and filter multiple forms of information quickly in a complex and dynamic environment, maintaining high levels of attention and efficiently allocating it, while keenly capturing and encoding external information. This process may repeatedly activate the DMN, VN, SMN, and FPN in cadets, ultimately altering the activation levels and functional connectivity of these networks in the resting state. Similarly, as shown in Fig 7 and Fig 8, it was found that among the high-contributing features obtained from the ML model, whether based on the BOLD signaling indicator or functional connectivity as feature inputs, there were consistently more features located in the left brain than in the right brain. This may be related to the focus of daily flight training. According to the theory of left-right brain dominance, the left brain is thought to excel at logical, rational, and computational tasks, while the right brain excels at artistic, creative, and spontaneous tasks [58]. However, the difference between them is relative rather than absolute.

In addition, for the ML models, we found that GNB and LR performed the best overall classification on this dataset. GNB and LR are suitable for data with simple linear relationships between features but are less effective for classifying high-dimensional complex data [59,60]. Therefore, we hypothesize that there may be simple linear associations between

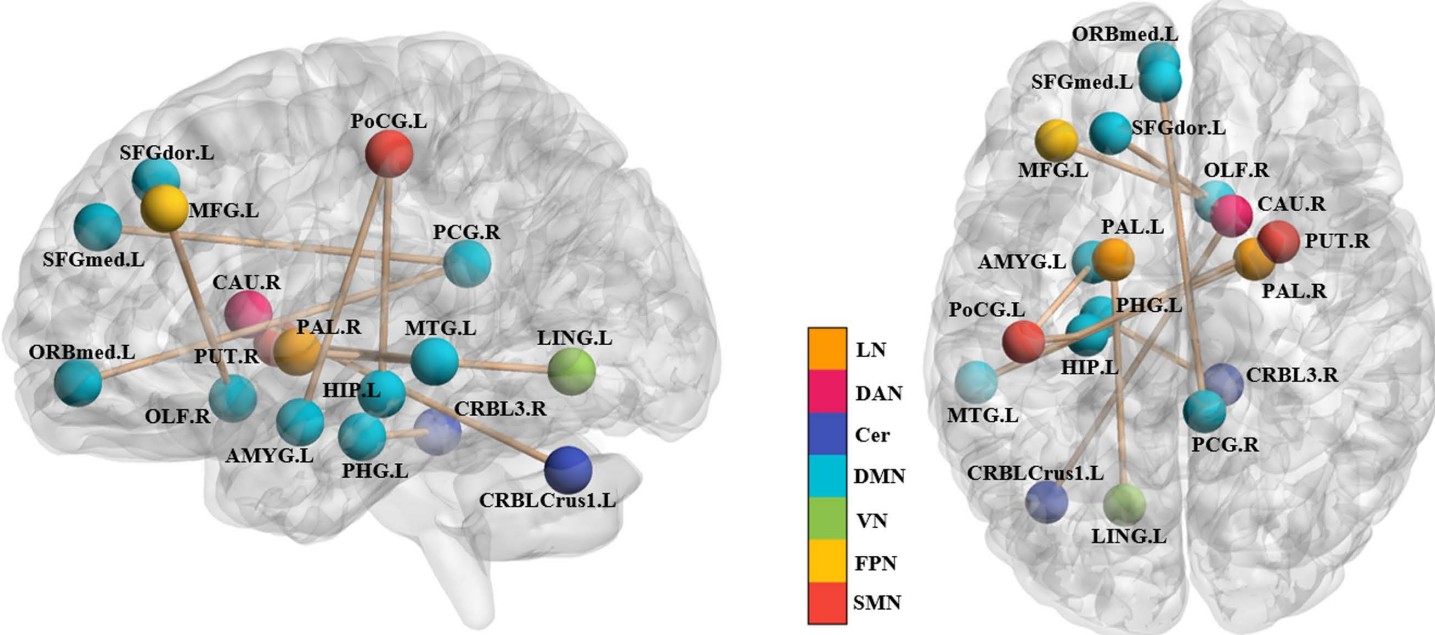

**Fig 8. Spatial distribution and associated brain networks of key connections in FC.** The various colors correspond to distinct cerebral networks.

the features screened based on the functional MRI data of flight cadets, but the specific association patterns need further investigation in combination with neurophysiological mechanisms.

The key brain regions identified in this study, along with the demonstrated effectiveness of the ML model in differentiating individuals who have received flight training, offer highly valuable reference for subsequent research. These research findings are expected to provide direction for constructing more accurate pilot prediction models. Furthermore, conducting an in – depth analysis of the brain networks where the key brain regions are located can help scientifically guide the curriculum arrangement of flight training and rationally plan the key content of training subjects according to the functional characteristics of each brain region. In this way, flight training can better conform to the development laws of brain functions, improve training effectiveness, and provide strong support for cultivating high – quality pilots.

There are still some shortcomings in this study. Our sample size is small, and to achieve better generalization ability, we need more participants and other modalities of neuroimaging data. Additionally, all participants were from the CAFUC and were limited to male right – handed cadets. Although this approach reduced the impact of potential confounding factors, the samples were sourced from the same region. Future research could consider incorporating more diverse samples.

## Conclusion

Our current study is based on resting-state fMRI data of flight cadets, from which four types of metrics were extracted. We developed and validated five machine learning models, finding that BOLD signal metrics exhibited the highest AUC, accuracy, sensitivity, and specificity, and were able to reach 0.93, 0.83, 0.85, and 0.82, respectively. The LR classifier demonstrated the best classification performance within this dataset. Modeling in conjunction with LR and BOLD metrics can effectively distinguish between flight cadets and ground cadets. We synthesized the BOLD signal and FC, revealing that key brain regions and connections with high feature weights were concentrated in audiovisual, motor, emotion control, and cognition-related areas. These regions, all from the DMN, VN, and SMN, showed high correlation with flight training,

suggesting that flight cadets' related brain networks were more active due to long-term training. This study combines fMRI techniques and machine learning methods to not only better identify differences between flight trainees and non-flight trainees, but also to reveal the effects of flight training on brain function. This research approach provides a neuroimaging perspective for future pilot training optimization, and helps to explore specific neural mechanisms in the pilot brain using machine learning methods.

## Author contributions

**Conceptualization:** Lu Ye.

**Formal analysis:** Xi Chen.

**Funding acquisition:** Xi Chen.

**Project administration:** Lu Ye, Dongfeng Yan.

**Software:** Shuhao Weng.

**Supervision:** Lu Ye, Dongfeng Yan, Shan Ma.

**Validation:** Shan Ma.

**Visualization:** Shuhao Weng.

**Writing – original draft:** Shuhao Weng.

**Writing – review & editing:** Lu Ye, Shuhao Weng, Dongfeng Yan.

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
