## [Decision Letter · Decision Letter 0]

5 Feb 2025

PONE-D-24-48304Recognition of flight cadets brain functional magnetic resonance imaging data based on machine learning analysisPLOS ONE

Dear Dr. Yan,

Thank you for submitting your manuscript to PLOS ONE. After careful consideration, we feel that it has merit but does not fully meet PLOS ONE’s publication criteria as it currently stands. Therefore, we invite you to submit a revised version of the manuscript that addresses the points raised during the review process. In particular, the reviewers raised concerns about the control group, and requested additional details in the methods and rationale for the study. 

We look forward to receiving your revised manuscript.

Kind regards,

Dzung Pham

Academic Editor

PLOS ONE

Journal requirements:   When submitting your revision, we need you to address these additional requirements. 1. Please ensure that your manuscript meets PLOS ONE's style requirements, including those for file naming. The PLOS ONE style templates can be found at https://journals.plos.org/plosone/s/file?id=wjVg/PLOSOne_formatting_sample_main_body.pdf and https://journals.plos.org/plosone/s/file?id=ba62/PLOSOne_formatting_sample_title_authors_affiliations.pdf. 2. Please note that PLOS ONE has specific guidelines on code sharing for submissions in which author-generated code underpins the findings in the manuscript. In these cases, we expect all author-generated code to be made available without restrictions upon publication of the work. Please review our guidelines at https://journals.plos.org/plosone/s/materials-and-software-sharing#loc-sharing-code and ensure that your code is shared in a way that follows best practice and facilitates reproducibility and reuse. 3. We note that the grant information you provided in the ‘Funding Information’ and ‘Financial Disclosure’ sections do not match.  When you resubmit, please ensure that you provide the correct grant numbers for the awards you received for your study in the ‘Funding Information’ section. 4. We note that you have indicated that there are restrictions to data sharing for this study. For studies involving human research participant data or other sensitive data, we encourage authors to share de-identified or anonymized data. However, when data cannot be publicly shared for ethical reasons, we allow authors to make their data sets available upon request. For information on unacceptable data access restrictions, please see http://journals.plos.org/plosone/s/data-availability#loc-unacceptable-data-access-restrictions.  Before we proceed with your manuscript, please address the following prompts: a) If there are ethical or legal restrictions on sharing a de-identified data set, please explain them in detail (e.g., data contain potentially identifying or sensitive patient information, data are owned by a third-party organization, etc.) and who has imposed them (e.g., a Research Ethics Committee or Institutional Review Board, etc.). Please also provide contact information for a data access committee, ethics committee, or other institutional body to which data requests may be sent. b) If there are no restrictions, please upload the minimal anonymized data set necessary to replicate your study findings to a stable, public repository and provide us with the relevant URLs, DOIs, or accession numbers. Please see http://www.bmj.com/content/340/bmj.c181.long for guidelines on how to de-identify and prepare clinical data for publication. For a list of recommended repositories, please see https://journals.plos.org/plosone/s/recommended-repositories. You also have the option of uploading the data as Supporting Information files, but we would recommend depositing data directly to a data repository if possible. Please update your Data Availability statement in the submission form accordingly. 5. In the online submission form, you indicated that [The data from this study are available upon request. However, the Civil Aviation Administration of China prohibits the public sharing of pilots’ physiological data due to legal constraints. Researchers interested in accessing these data may contact:yandongfeng@cafuc.edu.cn.]. All PLOS journals now require all data underlying the findings described in their manuscript to be freely available to other researchers, either 1. In a public repository, 2. Within the manuscript itself, or 3. Uploaded as supplementary information.This policy applies to all data except where public deposition would breach compliance with the protocol approved by your research ethics board. If your data cannot be made publicly available for ethical or legal reasons (e.g., public availability would compromise patient privacy), please explain your reasons on resubmission and your exemption request will be escalated for approval. 

Reviewers' comments:

Reviewer's Responses to Questions

**Comments to the Author**

1. Is the manuscript technically sound, and do the data support the conclusions?

Reviewer #1: Yes

Reviewer #2: Partly

2. Has the statistical analysis been performed appropriately and rigorously? 

Reviewer #1: I Don't Know

Reviewer #2: I Don't Know

3. Have the authors made all data underlying the findings in their manuscript fully available?

Reviewer #1: Yes

Reviewer #2: Yes

4. Is the manuscript presented in an intelligible fashion and written in standard English?

Reviewer #1: Yes

Reviewer #2: No

5. Review Comments to the Author

Reviewer #1: Overall, this is a very interesting manuscript that I would suggest is published with minor edits. The manuscript is well written with clear explanation and logic. I suggest minor edits for consideration below.

Materials and Methods

Please provide an explanation on the exposure of the control group. Was this group only involved in ground school training without any flight experience? Are these trainees in another carrier field? This will help the reader understand what is meant by ‘trainees from the ground program.’ I see that 0 hours of flight training was recorded in Table 2.

Table 2 – I am not clear what 39/0 and 37/0 mean in the Sex category. I would suggest reporting the n instead of the %, or put the % within parentheses.

On line 271, I might suggest adding “where feature is defined as…” to add clarity for the reader.

Line 295 – I would suggest changing the word ‘hue’ to ‘color.’

I would suggest Figure 7 and 8 (with explanation) be within the results section with interpretation in the discussion section.

The explanation of the BOLD signal was informative for the rest of the manuscript. I might recommend the authors provide a similar concise explanation of how the reader should conceptualize FC, ALFF, and ReHo as there is no explanations of what these functions are.

Lines 552 – 557:

Other limitations – only trainee pilots in one country with limited flying experience. I would suggest commenting on gender and handedness (see comment about Table 2). Additionally, each ML model had <100% sensitivity and specificity. I would suggest adding a line discussing how there are factors not identified in these models, which suggest they are incomplete.

I am not clear how any conclusions can be drawn from this project about how this ML model could be used to aid pilot selection. This study did not aim to identify who did/did not pass flight training. Instead, this study aimed to identify whether ML models can identify whether someone did or did not complete a certain degree of flight training. I would suggest removing this line. A line could be added about how these data could inform future working aiming to predict pilot performance in flight training/selection.

Line 569, I would recommend changing the word ‘indicating’ to ‘suggesting’

Line 571 – I would suggest another term besides ‘regular’ trainees

Line 572-576 – again, I am not sure how any conclusions can be drawn about ‘pilot screening and training optimization.’ I would suggest removing this line. This study demonstrated that a ML model can identify those who did and did not complete flight training and provides insight on neural connectivity. It would be reasonable suggest a line how future research could employ these data to support those objectives.

A line will need to be added stating whether the authors will provide their primary data upon request

Reviewer #2: The manuscript submitted by Le et al. examines the neural profile of flight cadets and how it differs from ground program cadets. All participants underwent a resting state fMRI scan which was used by the authors to extract four different metrics of brain activity (BOLD signal, functional connectivity, amplitude of low frequency fluctuation, and regional homogeneity). The authors developed five different classification models using machine learning to determine the best model(s) to analyze the resting state fMRI data. Overall, the BOLD signal was the best metric to classify flight cadets while the logistic regression and Gaussian Naïve Bayes models exhibited higher classification accuracy relative to other model types. Moreover, specific brain regions in the default mode, frontoparietal, somatomotor, and visual networks possess different connectivity patterns in flight cadets. In general, the manuscript examines an interesting topic that may guide future studies in personnel selection for certain occupations. However, there are a few points that the authors should address in the manuscript before acceptance can be considered:

Major:

1. Introduction (page 4, lines 67-68; page 5, lines 98-99): There are many terms that the authors mention but do not define or describe. Terms such as SN, DMN, CEN, ALFF, ReHo are all technical terminology that brain imaging researchers would understand, but for the average reader, which would likely be the case for a PLoS One audience, these terms should be defined. Some of these terms, such as DMN, are not defined until the Discussion (page 25, lines 434-439) when it would be more appropriate to define them in the Introduction instead. The aforementioned terms only require 1-2 sentences each to describe in the Introduction while a more detailed description of the relevant terms, such as DMN, can be elaborated on in the Discussion.

2. Introduction (page 5, lines 95-97): The study rationale seems incomplete. The authors do not elaborate why it is important to study brain function in healthy individuals. For example, an explanation of what constitutes flight training and how it is unique from other occupations (e.g., ground cadets) would be helpful. Similarly, the authors mention drug/alcohol use (lines 106-109) but it is unclear how this even relates to the study question.

3. Introduction (page 6, lines 110 – 119): This seems like an exploratory study, which should be made clear in the introduction. Furthermore, what were the authors’ hypotheses for the study questions/objectives?

4. Methods (page 8, 163-165): The authors removed the signal from cerebrospinal fluid, white matter, and head motion. It is common procedure for the resting state fMRI data to also remove physiological signals, such as heart beat and respiration. Could the authors explain why this was not collected and/or regressed out and how this would impact their resting state signal metrics?

5. Methods (Feature Extraction, page 8): The authors only explained how they calculated BOLD signal and FC. Where is the description for how ALFF and ReHo were computed?

6. Methods (Feature extraction, pages 8-10): Have the authors considered using graph theory as another resting state fMRI metric to be compared in the study? Graph theory allows resting state fMRI data to be analyzed with more granularity which is particularly relevant in this study when various metrics are being compared head-to-head [see Medaglia (2017) Neuroimaging Clin N Am, 27(4):593-607; Lv et al., (2018) AJNR Am J Neuroradiol, 39(8):1390-1399]. Given that the authors already have 116 brain regions outlined in Table 1 that they used in calculating the BOLD signal and FC, they can perform graph theory on these same regions to extract values such as clustering, path length, or efficiency.

7. Methods (page 9, lines 183-186) and Results (page 15, Table 3): It appears that the resting state fMRI metrics with the most features that have been extracted (29,000 for BOLD signal and 6,670 for FC) and selected (40 for BOLD signal and 141 for RC) are ultimately the features that will perform the best in the ML algorithms. While I recognize the authors applied more stringent p-value thresholds for the BOLD and RC metrics, the ALFF and ReHo metrics appear to already be at a disadvantage from the beginning as fewer features were extracted to be used in the ML model comparison and evaluation. In other words, how would the results differ if all 4 resting state fMRI metrics used the same number of features (e.g., only the top 10 features for each metric) for ML model comparison and evaluation?

8. Methods (pages 12-13, lines 221-223): The choice of the 5 classification models (GNB, SVM, RF, LR, and XGBoost) appears arbitrary. Please explain the rationale for these 5 model types over other machine learning models. Would the authors obtain similar results if they used k-nearest neighbor (KNN) or principal component analysis (PCA), as an example?

9. Discussion (pages 24-25, lines 416-425): The authors discuss the results from the perspective of neural changes due to cadet training. While this is a plausible explanation, this seems a bit misleading for this study since the authors did not compare flight cadets before and after a training course nor did they compare separate groups of flight cadets (experienced vs. novice). Instead, the authors compared a group of flight cadets versus ground cadets, the latter being a different occupation altogether. One cannot rule out the possibility that the neural profile for flight cadets is already different from ground cadets at the outset and may (or may not) be related to performance/success in flight cadet training, which of course requires further research to investigate.

Minor:

1. Introduction, page 5, line 106: Please add the word “functional” in front of magnetic resonance imaging.

2. Methods, page 7 (line 146) and page 8 (line 152): Please explicitly state the voxel size for the T1 and resting state fMRI scans. Are these 4 × 4 × 4 mm voxels?

3. Results, page 14, Table 2: Please clarify the number of participants in the flight cadet group. Table 2 says N = 26, but on page 14, line 243 it indicates that N = 39.

4. Discussion, page 26-27, lines 458-462: Both sentences seem redundant, please consider rewording or removing one of these sentences.

5. Please check for minor spelling typos throughout the manuscript (e.g., page 5, line 98 reads as: “… resting-state brain function, including Amplitude of ALFF, functional connectivity …” but either the words “Amplitude of” is not required or the letters “FF” should be spelled out).

6. PLOS authors have the option to publish the peer review history of their article (what does this mean? ). If published, this will include your full peer review and any attached files.

**Do you want your identity to be public for this peer review?** For information about this choice, including consent withdrawal, please see our Privacy Policy .

Reviewer #1: No

Reviewer #2: No

---

## [Author Response · Author response to Decision Letter 0]

17 Mar 2025

Dear Editor,

We are truly grateful to your positive comments and thoughtful suggestions on our manuscript (Manuscript Number: PONE-D-24-48304 entitled “Recognition of flight cadets brain functional magnetic resonance imaging data based on machine learning analysis”). Based on these comments and suggestions, we have made careful modifications on the original manuscript. All changes made to the revision are in light blue, and all changes used to revision in response letter marked with “****”. We hope the new manuscript will meet with your approval. Below you will find our point-by-point responses to your comments/ questions.

1) Text of editor’s comment: 1. Please ensure that your manuscript meets PLOS ONE's style requirements, including those for file naming. The PLOS ONE style templates can be found at https://journals.plos.org/plosone/s/file?id=wjVg/PLOSOne_formatting_sample_main_body

pdf and https://journals.plos.org/plosone/s/file?id=ba62/PLOSOne_formatting_sample_title_ authors_affiliations.pdf

Response: Thank you very much for your comment. We have revised the title page in accordance with the formatting requirements of PLOS ONE.

2) Text of editor’s comment: 2. Please note that PLOS ONE has specific guidelines on code sharing for submissions in which author-generated code underpins the findings in the manuscript. In these cases, we expect all author-generated code to be made available without restrictions upon publication of the work. Please review our guidelines at https://journals.plos.org/plosone/s/materials-and-software-sharing#loc-sharing-code and ensure that your code is shared in a way that follows best practice and facilitates reproducibility and reuse.

Response: Thank you very much for your comment. We have shared the available code as required and added the relevant statement in the revised manuscript.

Revision (lines 632-634):

“Codes and raw data related to this study can be found in the online repository: https://doi.org/10.6084/m9.figshare.28608278.v1”

3) Text of editor’s comment: 3. We note that the grant information you provided in the ‘Funding Information’ and ‘Financial Disclosure’ sections do not match. When you resubmit, please ensure that you provide the correct grant numbers for the awards you received for your study in the ‘Funding Information’ section.

Response: Thank you very much for your comment. The sources of funding for this study are as follows: the Sichuan Science and Technology Program (Grant No. 2023NSFSC1183) and the Fundamental Research Funds for the Central Universities (Grant No. J2023-004), with funding associated with Dongfeng Yan.

Revision (lines 628-630):

“This study was financially supported by Sichuan Science and Technology Program (Grant No. 2023NSFSC1183) and the Fundamental Research Funds for the Central Universities (Grant No. J2023-004).”

4) Text of editor’s comment: 4. We note that you have indicated that there are restrictions to data sharing for this study. For studies involving human research participant data or other sensitive data, we encourage authors to share de-identified or anonymized data. However, when data cannot be publicly shared for ethical reasons, we allow authors to make their data sets available upon request. For information on unacceptable data access restrictions, please see http://journals.plos.org/plosone/s/data-availability#loc-unacceptable-data-access-restrictions. Before we proceed with your manuscript, please address the following prompts:

Response: Thank you very much for your comment. We have carefully reviewed your guidance on data sharing and fully appreciate the journal's commitment to data transparency. In alignment with PLOS ONE's principles, we have revised the data access for this study to "all data are fully available without restriction" while ensuring compliance with ethical standards and relevant data protection laws.

Given the inclusion of sensitive personal information (e.g., pilots' physiological data), all datasets have undergone anonymization procedures to remove direct identifiers and prevent re-identification. The anonymized data will be made available via a public repository [https://doi.org/10.6084/m9.figshare.28608278.v1] to facilitate reproducibility and further research.

5) Text of editor’s comment: 5. In the online submission form, you indicated that [The data from this study are available upon request. However, the Civil Aviation Administration of China prohibits the public sharing of pilots’ physiological data due to legal constraints. Researchers interested in accessing these data may contact: yandongfeng@cafuc.edu.cn.].

Response: Thank you very much for your comment. We have carefully reviewed your guidance on data sharing and fully appreciate the journal's commitment to data transparency. In alignment with PLOS ONE's principles, we have revised the data access for this study to "all data are fully available without restriction" while ensuring compliance with ethical standards and relevant data protection laws.

Codes and raw data related to this study can be found in the online repository: https://doi.org/10.6084/m9.figshare.28608278.v1

II Manuscript Revision Overview

1. According to editor's comment 2) and 3), lines 632-634 and 628-630 have been revised.

2. According to reviewer 1’s comment 1), lines 156-162 have been revised.

3. According to reviewer 1’s comment 3), lines 323-324 have been revised.

4. According to reviewer 1’s comment 6), lines 220-230 and 212-215 have been revised.

5. According to reviewer 1’s comment 7), lines 604-607 have been revised.

6. According to reviewer 1’s comment 8), lines 592-601 have been revised.

7. According to reviewer 2’s comment 1), lines 64-77 have been revised.

8. According to reviewer 2’s comment 2), lines 104-118 have been revised.

9. According to reviewer 2’s comment 3), lines 129-134 have been revised.

10. According to reviewer 2’s comment 5), lines 220-230 have been revised.

11. According to reviewer 2’s comment 8), lines 270-277 have been revised.

12. According to reviewer 2’s comment 9), lines 150-156 have been revised.

All the spelling errors and omissions identified by Reviewer 1 and Reviewer 2 have been thoroughly corrected in the revised manuscript.

I Original comments of Reviewer #1:

Overall, this is a very interesting manuscript that I would suggest is published with minor edits. The manuscript is well written with clear explanation and logic. I suggest minor edits for consideration below.

Response to Reviewer #1:

Dear Reviewer,

We are truly grateful to your positive comments and thoughtful suggestions on our manuscript (Manuscript Number: PONE-D-24-48304 entitled “Recognition of flight cadets brain functional magnetic resonance imaging data based on machine learning analysis”). Based on these comments and suggestions, we have made careful modifications on the original manuscript. All changes made to the revision are in light blue, and all changes used to revision in response letter marked with “****”. We hope the new manuscript will meet with your approval. Below you will find our point-by-point responses to your comments/ questions.

1) Text of reviewer’s comment: 1. Please provide an explanation on the exposure of the control group. Was this group only involved in ground school training without any flight experience? Are these trainees in another carrier field? This will help the reader understand what is meant by ‘trainees from the ground program.’ I see that 0 hours of flight training was recorded in Table 2.

Response: Thank you very much for your comments. We will provide a more detailed description of the background of the control group in the "Participants" section to help readers better understand the differences between the two groups.

Revision (lines 156-162):

“The control group consisted of cadets who received no flight training and only participated in ground-based theoretical courses (such as aviation theory, meteorology, and navigation principles). All ground-based cadets came from the engineering discipline of the same institution, and possessed neither a flight license nor any experience in flight simulation. Their training content significantly differed from that of the flight cadets, focusing primarily on theoretical learning and ground operation skills, rather than practical flight tasks.”

2) Text of reviewer’s comment: 2. Table 2 – I am not clear what 39/0 and 37/0 mean in the Sex category. I would suggest reporting the n instead of the %, or put the % within parentheses.

Response: We greatly appreciate your identification of the issue. In accordance with your suggestion, we have revised the data presentation in the gender column of Table 2 to: Gender (% male), and the specific values for both the flight cadets and the control group in this column have been updated to 100%.

3) Text of reviewer’s comment: 3. On line 271, I might suggest adding “where feature is defined as…” to add clarity for the reader.

Response: We sincerely appreciate your suggestion. We will include an additional explanation of the term "feature" in the revised manuscript to provide readers with a clearer understanding of this concept.

Revision (lines 323-324):

“In this study, features are defined as quantitative metrics extracted from resting-state fMRI data, including BOLD signals, FC, ALFF, and ReHo.”

4) Text of reviewer’s comment: 4. Line 295 – I would suggest changing the word ‘hue’ to ‘color.’

Response: Thank you very much for your suggestion. In the revised manuscript, we have changed "hue" to "color."

5) Text of reviewer’s comment: 5. I would suggest Figure 7 and 8 (with explanation) be within the results section with interpretation in the discussion section.

Response: Thank you very much for your suggestion. Regarding your suggestion to move Figure 7 and Figure 8, along with their explanations, to the Results section and then interpret them in the Discussion section, we understand that you aim to make the article's structure clearer and its logic more coherent through this adjustment. However, in our current article framework, these two figures serve as visual representations of the key findings in the research results. Their main purpose is to display the key brain regions and connections identified based on different indicators (BOLD signals and FC). In the Results section, we primarily focus on presenting the classification performance at the data level (such as AUC, sensitivity/specificity) and the screening results of key features (Table 3, Figure 1 - 4). In the Discussion section, we conduct in - depth discussions from perspectives such as neuroscience theories and the actual impact of flight training, in combination with these findings. Placing Figure 7 and Figure 8 in the Discussion section can more effectively help readers visually observe the positions and distributions of the relevant brain regions and connections while understanding the association between theory and practice, thus enabling them to better comprehend our research conclusions. Therefore, considering the integrity and logic of the article, we hope to retain the current positions of the figures.

6) Text of reviewer’s comment: 6. The explanation of the BOLD signal was informative for the rest of the manuscript. I might recommend the authors provide a similar concise explanation of how the reader should conceptualize FC, ALFF, and ReHo as there is no explanations of what these functions are.

Response: Thank you very much for your suggestion. In our manuscript, we indeed did not provide detailed explanations for these indicators, which may cause inconvenience to readers. We will briefly introduce FC, ALFF, and ReHo in the "Feature Extraction" section. We will illustrate their roles in brain function research and explain how to calculate these indicators from fMRI data.

Revision (lines 220-230, lines 212-215):

“In this study, the calculation of ALFF involves performing a Fourier transform on the time series of each voxel to obtain the power spectrum within the low-frequency range (0.01 - 0.08 Hz). The square root of this power spectrum is then computed to derive the ALFF value for each voxel. This value effectively reflects the intensity of spontaneous neural activity in the brain and provides a critical metric for assessing the brain's intrinsic functional state. ReHo, on the other hand, is computed using Kendall’s coefficient of concordance (KCC). Specifically, the KCC value is calculated by comparing the time series of each voxel with those of its neighboring voxels, which measures the temporal coherence within local brain regions. This metric is crucial for understanding the synchrony of neural activity within localized brain areas, contributing to the exploration of the characteristics of local brain functional integration.”

“FC can reflect the degree of connection tightness between various brain regions in the brain functional network, which helps to understand how different regions of the brain cooperate with each other functionally.”

7) Text of reviewer’s comment: 7. Lines 552 – 557: Other limitations – only trainee pilots in one country with limited flying experience. I would suggest commenting on gender and handedness (see comment about Table 2). Additionally, each ML model had <100% sensitivity and specificity. I would suggest adding a line discussing how there are factors not identified in these models, which suggest they are incomplete.

Response: Thank you very much for your comments. We will supplement the description of limitations and comments on gender and handedness in the revised manuscript according to your suggestions. Regarding the issue you raised that the sensitivity and specificity of each machine - learning model are less than 100%, we are well aware of your concern about the accuracy and integrity of the models. However, in the field of machine learning, due to various factors such as data complexity and the limitations of the models themselves, especially considering that fMRI data is more difficult to reach a high level due to individual differences, measurement noise, and environmental factors. In all the studies related to fMRI combined with machine learning referenced in this research, the evaluation metrics for the machine learning models did not reach 100%. From this perspective, the failure of the model performance to reach 100% is a normal phenomenon in this field and is not a problem unique to this study.

Furthermore, the core goal of the machine - learning field is to discover poten

---

## [Decision Letter · Decision Letter 1]

21 Apr 2025

Recognition of flight cadets brain functional magnetic resonance imaging data based on machine learning analysis

PONE-D-24-48304R1

Dear Dr. Yan,

We’re pleased to inform you that your manuscript has been judged scientifically suitable for publication and will be formally accepted for publication once it meets all outstanding technical requirements.

Kind regards,

Dzung Pham

Academic Editor

PLOS ONE

Additional Editor Comments (optional):

Reviewers' comments:

Reviewer's Responses to Questions

**Comments to the Author**

1. If the authors have adequately addressed your comments raised in a previous round of review and you feel that this manuscript is now acceptable for publication, you may indicate that here to bypass the “Comments to the Author” section, enter your conflict of interest statement in the “Confidential to Editor” section, and submit your "Accept" recommendation.

Reviewer #2: All comments have been addressed

2. Is the manuscript technically sound, and do the data support the conclusions?

Reviewer #2: Yes

3. Has the statistical analysis been performed appropriately and rigorously? 

Reviewer #2: Yes

4. Have the authors made all data underlying the findings in their manuscript fully available?

Reviewer #2: Yes

5. Is the manuscript presented in an intelligible fashion and written in standard English?

Reviewer #2: Yes

6. Review Comments to the Author

Reviewer #2: The authors have done an admirable job in addressing my comments. The manuscript is much improved with a complete study rationale, definition of terms at the first instance, and clear hypothesis, all of which help the reader in understanding the logic behind the study. I also appreciate that the authors conducted a supplementary analysis to further validate that BOLD and FC indicators perform the best in ML algorithms. I have no additional concerns and recommend this manuscript for publication.

7. PLOS authors have the option to publish the peer review history of their article (what does this mean? ). If published, this will include your full peer review and any attached files.

**Do you want your identity to be public for this peer review?** For information about this choice, including consent withdrawal, please see our Privacy Policy .

Reviewer #2: No

---

## [Editor Report · Acceptance letter]

PONE-D-24-48304R1

PLOS ONE

Dear Dr. Yan,

I'm pleased to inform you that your manuscript has been deemed suitable for publication in PLOS ONE. Congratulations! Your manuscript is now being handed over to our production team.

Kind regards,

on behalf of

Dr Dzung Pham

Academic Editor

PLOS ONE